# Extra-Capsular Spread of Lymph Node Metastasis in Oral, Oropharyngeal and Hypopharyngeal Cancer: A Comparative Subsite Analysis

**DOI:** 10.3390/cancers16030659

**Published:** 2024-02-03

**Authors:** Yung Jee Kang, Goeun Park, Sung Yool Park, Taehwan Kim, Eunhye Kim, Yujin Heo, Changhee Lee, Han-Sin Jeong

**Affiliations:** 1Department of Otorhinolaryngology-Head and Neck Surgery, Samsung Medical Center, Sungkyunkwan University School of Medicine, Seoul 06351, Republic of Korea; yungjee.kang@samsung.com (Y.J.K.); sungyool.park@samsung.com (S.Y.P.); foreneo019@gmail.com (T.K.); eh0502.kim@samsung.com (E.K.); yujin02.heo@samsung.com (Y.H.); changhee4559.lee@samsung.com (C.L.); 2Center for Biomedical Statistics, Samsung Medical Center, Seoul 06351, Republic of Korea; gogoeun.park@samsung.com

**Keywords:** head and neck cancer, oral tongue cancer, pharynx cancer, lymph node metastasis, extra-capsular spread

## Abstract

**Simple Summary:**

The extra-capsular spread (ECS) of lymph node metastasis (LNM) is a hallmark of aggressive tumor phenotype in head and neck cancer (HNC) and an important adverse prognostic factor for recurrence, metastasis, and patient survival. According to the previous studies, the LNM characteristics of HNC (i.e., the growth and spread of LNM) well correlate with the ECS occurrence, but there has been a disagreement about the primary tumor characteristics and ECS. Thus, we investigated the risk factors of ECS across different HNC subsites comparatively. In this study, we confirmed that LNM variables were significant risk factors for ECS in all subsites of HNC. Interestingly, in oral tongue cancer, tumor dimensional variables were significantly related to ECS; however, in oro- and hypopharyngeal cancer, neither the primary tumor dimension nor the T status were significant factors for ECS occurrence. Site-dependent primary tumor characteristics or nodal status might contribute differentially to the ECS development of LNM in HNC.

**Abstract:**

Background: The extra-capsular spread (ECS) of lymph node metastasis (LNM) is a hallmark of aggressive primary tumor phenotype in head and neck cancer (HNC); however, the factors influencing ECS are poorly understood. Patients and Methods: This was a retrospective study, including 190 cases of oral tongue cancer (OTC), 148 cases of oropharyngeal cancer (OPC) (118 HPV-positive and 30 HPV-negative), and 100 cases of hypopharyngeal cancer (HPC). Tumor dimension, tumor biological variables (lymphovascular/perineural invasion and histologic grade), and LNM variables (LNM number and size) were analyzed according to the presence of ECS using multivariable logistic regression and receiver operating characteristic (ROC) curve analyses. Results: LNM variables were significant factors for ECS in all subsites of HNC (*p* < 0.05), except HPV-positive OPC. In OTC, tumor dimensional variables were significantly related to ECS (*p* < 0.01). Meanwhile, in OPC and HPC, neither the primary tumor dimension nor the T status were significant factors for ECS occurrence. The predictability of ECS by ROC curve using multiple variables was 0.819 [95% confidence interval: 0.759–0.878] in OTC, 0.687 [0.559–0.815] in HPV-positive OPC, 0.823 [0.642–1.000] in HPV-negative OPC, and 0.907 [0.841–0.973] in HPC. Conclusion: LNM variables were correlated with ECS occurrence for most HNC subsites, and site-dependent primary tumor characteristics might contribute differentially to the ECS development of LNM in HNC.

## 1. Introduction 

The extra-capsular spread (ECS) or extra-nodal extension of lymph node metastasis (LNM) is defined as tumor overgrowth beyond the lymph node capsule into the surrounding tissues [1]. The presence of ECS is a significant prognostic indicator of worse treatment outcomes and patient survival across many types of cancer [2,3,4,5,6]. In head and neck cancer (HNC), the ECS of LNM is regarded as an important adverse prognostic factor for local recurrence, distant metastasis, and patient survival [7,8,9,10,11,12,13,14,15,16,17]. 

Despite the overall negative impact of ECS in HNC, the site-specific effect of ECS has been reported to vary in HNC depending on the primary tumor site [10,11,18,19,20]. A study reported that ECS worsened disease-free survival in oral cavity cancer but not in human papillomavirus (HPV)-positive or -negative oropharyngeal malignancy [18]. Meanwhile, other studies suggested that ECS significantly diminished the overall survival rates of HPV-positive cancer patients [21,22,23,24,25,26]. Primary tumor sites in HNC, such as the hypopharynx, oropharynx, and larynx, can differentially affect the occurrence of ECS, with an ECS incidence of 70%, 37%, and 33% reported for these locations, respectively [27,28]. 

In addition to the primary tumor sites in HNC, studies have shown that multiple other factors also contribute to the development of ECS in LNM. Focusing on LNM characteristics, the number and size of LNM are good predictors of ECS [29]. Clinical N classification also has a strong correlation with ECS [27,30,31], and ECS has been reported in 60–100% of LNM larger than 3 cm [27,32,33]. However, the incidence of ECS seemed not to increase linearly with the size of LNM [34].

The primary tumor characteristics, including the depth of invasion (more than 10 mm), the pattern of invasion (tumor budding), and the characteristics of the tumor microenvironment (desmoplastic reaction or tumor-infiltrating lymphocytes) have been reported to be independent indicators for the ECS of LNM in oral squamous cell carcinoma [35]. We demonstrated in a previous study that the tumor dimension factor is a stronger predictor for LNM in oral tongue cancer (OTC) than in oropharyngeal cancer (OPC) and hypopharyngeal cancer (HPC) [36].

However, there is a lack of consistency in the findings of published studies, particularly on the primary tumor characteristics [37]. A study revealed that the pattern of invasion in the primary tumors such as the worst pattern of invasion (WPOI) was not correlated with the presence of ECS in OTC [38]. In another study, primary tumor size, primary site, histologic grade, and depth of invasion of tumor did not show significant associations with the ECS of LNM [33]. However, tumor dimension and perineural invasion were reported as independent risk factors for ECS in cT1-2N0 oral cavity squamous cell carcinomas [39]. To summarize the results of previous studies, LNM characteristics of HNC (i.e., the growth and spread of LNM) well correlate with the ECS occurrence, but there has been a disagreement about the primary tumor characteristics and ECS according to the tumor site.

Given the inconsistency among findings, we were motivated to reinvestigate the significant determinants or predictors of ECS across different HNC subsites comparatively and comprehensively. ECS is a hallmark of an aggressive primary tumor phenotype, and accurately predicting ECS can be essential for determining treatment options in HNC [10,27,40,41,42,43]. Therefore, in this study, we investigated risk factors of ECS occurrence in OTC, HPV-positive and HPV-negative OPC, and HPC.

## 2. Materials and Methods

### 2.1. Study Patients

This study was a retrospective single-center study. Inclusion criteria were patients with pathologically confirmed HNC who underwent primary surgical treatment at our hospital from 1996 to 2021. Among HNC subsites, we selected only cases of pathologically proven squamous cell carcinomas arising from the oral tongue, oropharynx, and hypopharynx, to minimize overlapping lesions.

The initial number of cases collected was 1542 comprising 778 OTC, 470 OPC, and 294 HPC cases. Patients without sufficient clinical and pathological information were excluded, leaving 863 cases, consisting of 570 cases of OTC, 176 cases of OPC (138 HPV-positive and 38 HPV-negative cases), and 117 cases of HPC. Neoadjuvant therapies before surgery were not performed in all patients. Patients with adverse features in surgical pathology received adjuvant therapies after surgery (adjuvant radiation therapy, RT = 168, 19.5%, adjuvant chemoradiation, CCRT = 336, 38.9%) (RT 18.2%, CCRT 25.6% in OTC; RT 29.0%, CCRT 60.9% in HPV-positive OPC; RT 23.7%, CCRT 52.6% in HPV-negative OPC; RT 12.8%, CCRT 73.5% in HPC). Among the 863 cases, we only included N(+) cases (*n* = 438) for our ECS analyses, excluding 425 N0 cases. Our study was approved by the Institutional Review Board of our institution (No. 2023-02-141).

### 2.2. Histopathology Analysis

Pathological diagnosis was confirmed by two pathologists with more than 5 years of experience in head and neck cancer pathology. Any ambiguity in the diagnosis was resolved by a joint discussion. In this study, we compared the T and N status according to the presence or absence of ECS; therefore, pathological tumor staging was based on the American Joint Commission on Cancer (AJCC) Cancer Staging Manual, 7th edition, where ECS was not included in the staging criteria [13,14]. 

For the included cases, the following data were collected: the presence of ECS, the primary tumor size, histologic grade, depth of invasion (DOI), the worst pattern of invasion (WPOI) [44], the presence of perineural invasion (PNI) or lymphovascular invasion (LVI), the status of p16 immunohistochemistry, LNM number, and the largest LNM size (diameter). Among the tumor biological variables, WPOI data were only available for OTC, and our institute did not routinely examine WPOI in other HNCs. Primary tumor volume was calculated as follows: 1/2 × [long axis] × [short axis] × [tumor thickness]. In OPC, we defined T0 as p16-positive or HPV-positive LNM, but no primary tumor was identified in the surgical pathology of the oropharynx.

### 2.3. Outcome Measurement

We analyzed the potential variables associated with the presence of ECS based on the primary tumor site. We classified variables into three categories, namely primary tumor dimensional (physical) variables, tumor biological variables, and LNM variables, according to the previous reports [27,29,32,33,35,36]. 

Tumor dimensional variables were tumor diameter, volume, depth of invasion, and pT classification. Tumor biological variables were LVI, PNI, WPOI, and tumor histologic grade (well, moderately, or poorly differentiated). LNM variables were the number of metastatic lymph nodes, the largest LNM size, and pN classification. 

### 2.4. Statistical Analysis

We examined whether tumor dimensional variables, tumor biologic variables, and LNM variables were associated with the presence of ECS for each tumor site. Wilcoxon rank-sum test for continuous variables and chi-square test or Fisher’s exact test for categorical variables were used to assess the significance of differences in variables according to ECS positivity versus negativity status. The frequency of ECS positivity according to the T and N status was evaluated with the Cochran–Armitage trend test.

Univariable and multivariable logistic regression analyses were used to identify the risk factors of ECS according to the primary tumor subsite. We designed multivariable models from the univariable analyses using a stepwise variable selection method. Receiver operating characteristics (ROC) curve analyses were used to estimate the predictability of ECS based on the physical and biological characteristics of the tumor and LNM variables for each tumor subsite, and the area under the curve (AUC) was calculated and compared. AUC comparison was conducted using the bootstrap method. Leave-one-out cross-validation (LOOCV) was used for the internal validation of the final model for each tumor subsite. We performed statistical analyses using SAS version 9.4 (SAS Institute, Cary, NC, USA) and R package version 4.2.1 (The Comprehensive R Archive Network, http://www.R-project.org, accessed on 1 April 2023.). A *p*-value < 0.05 was considered statistically significant.

## 3. Results

### 3.1. Clinical and Pathological Differences According to the Presence of ECS and Tumor Subsite

The baseline characteristics of our study cohort are presented in Table 1. Overall, there were 59.5%, 61.0%, 56.7%, and 81.0% of ECS-positive cases for OTC, HPV-positive OPC, HPV-negative OPC, and HPC tumor subsites among N(+) diseases, respectively. Age distribution was similar across tumor subsites and according to ECS positivity. Male predominance (>85%) was noted in all HNC subsites, with males slightly outnumbering females among OTC patients.

We classified several variables into three categories: tumor dimensional variables, tumor biological variables, and LNM variables (see Section 2.3). In OTC, the tumor dimension, DOI, T status, LVI, PNI, histologic grade, N status, the number of LNM, and the size of the largest LNM were significantly different between ECS-positive and ECS-negative cases. In HPV-negative OPC, a higher T status was associated with ECS occurrence. LNM variables were significantly different between cases with and without ECS for all subsites of HNC.

To confirm these findings, we conducted a trend test (linear-by-linear association) of ECS positivity according to the T and N status (Figure 1). In concordance with the previous results, the frequency of ECS-positive cases increased linearly with the T status in OTC and HPV-negative OPC but not in HPV-positive OPC and HPC. Meanwhile, ECS positivity increased proportionally to the N status for all subsites of HNC.

### 3.2. Univariable and Multivariable Analyses of Risk Factors for ECS

To determine the risk factors for ECS according to each tumor subsite, we performed univariable and multivariable analyses of potential variables. In OTC, univariable analyses revealed that tumor dimensional variables (tumor diameter, tumor volume, DOI > 10 mm, and T status) were significantly associated with ECS (Table 2). As tumor size was closely associated with the T status, we constructed a multivariable model with a variable selection method, in which only the T status among the tumor dimensional variables remained a significant risk factor for ECS in OTC. In addition, LNM variables such as the largest node size and the N status were also significantly associated with ECS occurrence in OTC.

In the case of HPV-positive OPC, no tumor dimensional variables were risk factors for ECS, while the largest node size of LNM was selected for a multivariable model with a marginal significance (*p* = 0.062). Among the tumor biological variables, LVI was a risk factor for ECS in HPV-positive OPC (*p* < 0.05) (Table 3).

In HPV-negative OPC, a higher T status was associated with a higher incidence of ECS in univariable analysis (Table 1 and Table 4, Figure 1). However, in the multivariable model, no tumor dimensional variables were risk factors of ECS for HPV-negative OPC. The N status was significantly associated with ECS in HPV-negative OPC (Table 4). As for HPC, no tumor dimensional variables were risk factors for ECS in a multivariable model, similar to HPV-positive and HPV-negative OPC (Table 5). Among the LNM variables, the size of the largest LNM and the N status were risk factors for ECS in HPC.

### 3.3. Predictability of ECS by ROC Curve Analyses

To evaluate the predictability of ECS based on tumor dimensional and biological characteristics and LNM variables for each tumor subsite, ROC curve analyses were conducted. Tumor dimensional and biological characteristics and LNM variables predicted ECS in OTC with AUC values of 0.672 (95% confidence interval, 95% CI = 0.600–0.744), 0.587 (95% CI = 0.518–0.657), and 0.788 (95% CI = 0.724–0.852), respectively. When all variables were combined, the AUC value was 0.819 (95% CI = 0.759–0.878). AUC values were statistically different among multivariable models (*p* < 0.001); however, the difference in AUC values was minimal between the models including LNM variables and all variables combined (*p* = 0.049) (Figure 2). The internal validation of the final model containing all variables was conducted using the LOOCV method. An AUC value of 0.780 (95% CI = 0.725–0.854) was calculated for ECS in OTC using the model with all variables combined.

In contrast to OTC, only LNM variables remained significantly in the final multivariable models for HPV-positive OPC, HPV-negative OPC, and HPC (Table 3, Table 4 and Table 5). In HPV-positive OPC, the models containing a tumor biological variable (LVI) and LNM variable (size of the largest LNM) predicted ECS with AUC values of 0.578 (95% CI = 0.506–0.651) and 0.647 (95% CI = 0.514–0.779), respectively. The difference in AUC values of the model containing the LNM variables and all variables combined (0.687, 95% CI = 0.559–0.815) was not statistically significant (*p* = 0.467) (Figure 2). Internal validation was conducted for LNM variables and all variables combined in HPV-positive OPC as neither of the two models showed statistical significance. An AUC value of 0.717 (95% CI = 0.511–0.723) was calculated for LNM variables, and an AUC value of 0.640 (95% CI = 0.528–0.752) was calculated for the model containing all variables combined.

In HPV-negative OPC, the models including tumor dimensional variables, LNM variables, and all variables combined predicted ECS with AUC values of 0.835 (95% CI = 0.714–0.956), 0.821 (95% CI = 0.696–0.946), and 0.823 (95% CI = 0.642–1.000), respectively. There were no significant differences among the three models (*p* > 0.05). As there was no difference among these models, we selected the model containing LNM variables as the final model because of the clinical significance of these variables. The internal validation of the final model revealed an AUC value of 0.701 (95% CI = 0.489–914) in HPV-negative OPC. In HPC, the models containing tumor biological variables and LNM variables predicted ECS with AUC values of 0.684 (95% CI = 0.599–0.769) and 0.907 (95% CI = 0.841–0.973), respectively. The model containing all variables combined was identical to the model of LNM variables only for HPC, which confirmed the importance of the LNM variables in predicting ECS occurrence in HPC. The internal validation of the HPC model with LNM variables yielded an AUC value of 0.884 (95% CI = 0.809–0.959). 

## 4. Discussion

ECS, which is defined as cancer cell extension beyond the lymph node capsule, is a major predictor of recurrence, metastasis, and poor prognosis in HNC patients, and treatment should be intensified in patients with ECS-positive tumors [1,41]. Therefore, understanding the risk factors and sequence of ECS occurrence may help guide treatment options and predict prognosis in HNC patients [4,10]. Although several studies have investigated the risk factors of ECS occurrence in HNC, findings have been inconsistent results among studies. The number and size of metastatic nodes and the depth and pattern of invasion were reported to be independent predictors for the ECS of LNM in HNC [17,27,29,30,31,32,33,35,45]. However, another study found that tumor size, location, histologic grade, and DOI were not risk factors for the ECS of LNM in oral cavity cancer [40]. These conflicting results led us to investigate the potential predictors of the ECS of LNM according to the HNC subsite. 

We previously demonstrated that the development of LNM in OTC is primarily dependent on tumor dimensions, while LNM in OPC and HPC is more influenced by biological factors [36]. As an extension of these findings, we found here that the ECS of LNM is highly dependent on tumor dimensional variables in OTC. LNM variables such as the number or the largest size of LNM or the N status were highly correlated with ECS occurrence for all subsites of HNC. Therefore, we conclude that the site-dependent primary tumor characteristics and nodal status contribute differentially to the development of the ECS of LNM in HNC. 

The mechanisms underlying ECS development have been investigated in several studies, but the relative significance of dimensional (physical) versus biological factors remains unclear. Many previous studies have focused on the differential expression of various genetic, epigenetic, or molecular markers in primary tumors with and without ECS [31,46,47,48,49,50,51,52,53], but only a few studies have directly compared ECS tumors with intra-nodal metastasis [54,55,56,57]. A strong correlation between the ECS of LNM and poor clinical outcomes in cancer patients can be explained by the aggressive phenotype of the primary tumor, and ECS findings may be just one manifestation of this aggressive phenotype [4]. Indeed, the hypothesis that the molecular determinants of ECS are present in the primary tumor is supported by recent studies [58,59,60,61]. Meanwhile, several studies have investigated the role of tumor microenvironment in ECS tumor formation [50,57,62,63,64] and suggested that phenotypic changes occur from intranodal metastasis to ECS, rather than the intrinsic properties of the primary tumor. 

The starting point for this study was the simple clinical observation that the frequency of ECS positivity increased with the N status in our HNC patients (Figure 1). If the molecular determinants of ECS are solely dependent on the primary tumor characteristics, the frequency of ECS will be similar (evenly distributed) regardless of the burden of LNM or reciprocal (early manifestation of ECS even in N1 disease). In contrast, we demonstrated that ECS occurrence is related to LNM progression in this study. Although this does not completely exclude the possibility that ECS determinants are already present in the primary tumor, molecular changes during LNM progression could be critical for ECS formation and dissemination, in addition to inherent primary tumor factors. Therefore, therapeutic interventions for ECS-prone changes may be a promising treatment option for improving the dismal ECS-associated outcomes in HNC patients.

Another notable finding of this study is that tumor dimensional (physical) factors were one of the determinants of ECS occurrence in OTC but not OPC or HPC. This finding can be explained by our previous study results; we demonstrated that the development of LNM in OTC was mainly dependent on tumor dimension [36]. Consequently, the progression of LNM appears to contribute to the development of ECS in OTC. Meanwhile, ECS and LNM could be independent of primary tumor dimension factors in OPC and HPC, which is supported by the clinical finding that many unknown primary tumors with neck lymph node metastasis originate from hidden OPC and HPC [65,66,67]. 

Although tumor dimensional and LNM variables were good predictors of ECS in OTC cases with an AUC of 0.672–0.788, the model containing all variables combined was the best predictor of ECS with an AUC over 0.8, indicating that dimensional (physical) and biological features of the tumor in addition to LNM variables contribute synergistically to ECS in OTC. LNM variables showed higher predictability for ECS in HPV-positive OPC and HPC with AUC values of 0.647 and 0.907, respectively. In HPV-positive OPC, LVI was a risk factor for ECS in the multivariable model. Biological variables (LVI) might affect ECS development because the model with all variables combined (AUC of 0.687) and LNM variables (AUC of 0.647) both predicted ECS similarly. Though the predictive model based on tumor dimensional variables had an AUC of 0.835 in HPV-negative OPC (which was not a significantly different AUC compared with the AUC of other models), this could be due to the small sample size of HPV-negative OPC cases. We selected the model containing LNM variables as the final model for HPV-negative OPC, considering the clinical significance. Our results should be confirmed in a large cohort study of HNC. 

This study had several limitations. First, this was a retrospective study conducted at a single institution based on postoperative pathologic data [37]. Therefore, there could have been selection bias due to the retrospective nature of the study. In addition, our data cannot inform treatment decisions because preoperative data or radiological tools were not analyzed [68]. Second, only a few OPC patients were enrolled in this study, limiting our ability to identify significant predictors of ECS in these patients. Third, we did not analyze some biomarkers and mechanisms reported to be predictors of ECS in earlier studies. Rather, we simply included pathological findings, which were readily available, under the category of tumor biological factors. Tumor biology dictated by specific molecular markers should be explored further in future studies. Lastly, we did not incorporate the extent of ECS into our analysis [69] but rather dichotomized ECS status as either present or absent. The extent of ECS is known to affect treatment outcomes and patient prognosis in HNC [12,70,71,72]. Therefore, a study investigating potential factors predicting ECS that include the extent of ECS could provide further insights into ECS at HNC subsites.

## 5. Conclusions

This was the first study to identify the potential risk factors for determining ECS occurrence according to HNC subsite, comparatively. The LNM characteristics of HNC (i.e., the growth and spread of LNM) well correlate with the ECS occurrence, and it appears that ECS is a secondary phenomenon that occurs during LNM progression in HNC. In particular, ECS and LNM depend on tumor physical dimensions in OTC. By contrast, ECS occurrence seems to be independent of primary tumor dimensions in OPC and HPC.

## Figures and Tables

**Figure 1 cancers-16-00659-f001:**
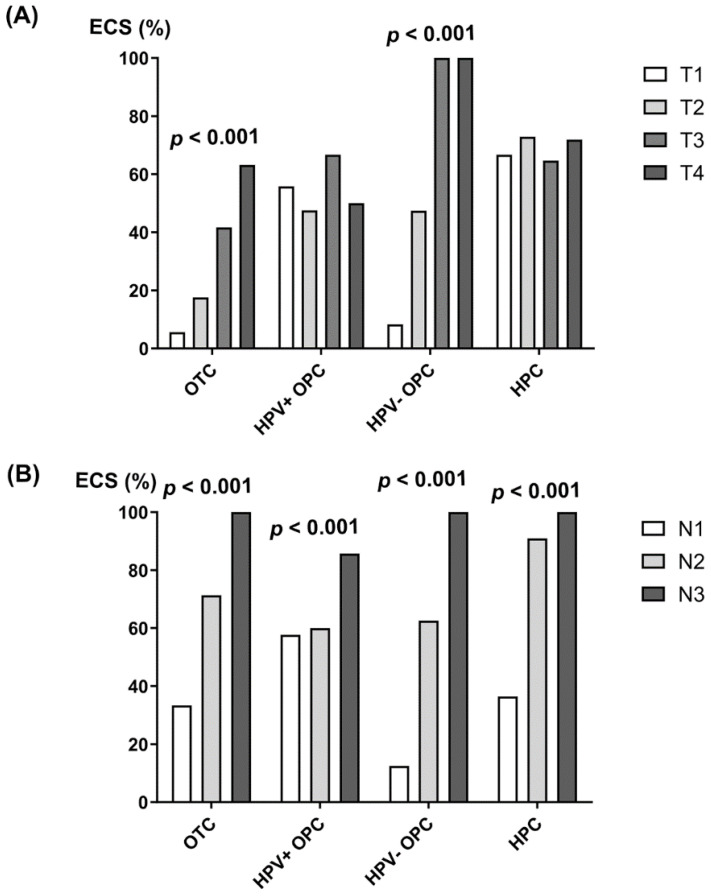
ECS percentage according to (**A**) T and (**B**) N status in OTC, HPV-positive and HPV-negative OPC, and HPC; *p*-values were calculated by the Cochran–Armitage trend test.

**Figure 2 cancers-16-00659-f002:**
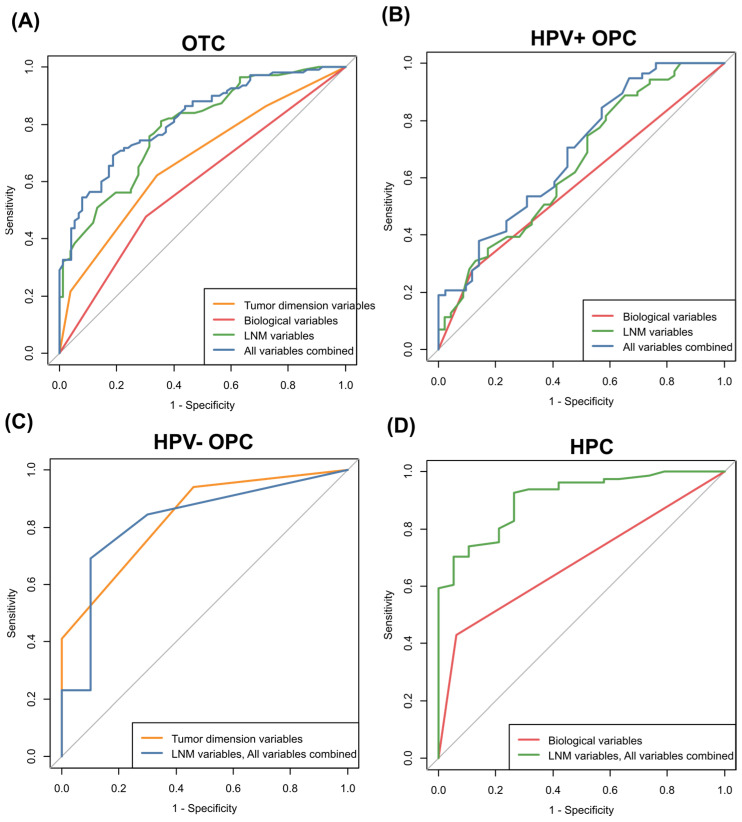
Receiver operating characteristics (ROC) curve analyses to estimate the predictability of ECS according to tumor physical, biological, and LNM variables at each tumor subsite. Areas under the curves (AUCs) were compared among the models: (**A**) OTC: AUC of a model using tumor dimensional variables = 0.672, AUC of a model using biological variables = 0.587, AUC of a model using LNM variables = 0.788, AUC of a model using all variables combined = 0.819; *p*-value for the difference between models was <0.001, except for the *p*-value of LNM variables and all variables (0.049). (**B**) HPV-positive OPC: AUC of a model using tumor biological variables = 0.578, AUC of a model using LNM variables = 0.647, AUC of a model using all variables combined = 0.687; *p*-values for the difference between models were >0.05. (**C**) HPV-negative OPC: AUC of a model using tumor dimension variables = 0.835, AUC of a model using LNM variables = 0.821, AUC of a model using all variables combined = 0.823; *p*-values for the difference between models were >0.05. (**D**) HPC: AUC of a model using biological variables = 0.684, AUC of a model using LNM variables = AUC of a model using all variables combined = 0.907; *p*-value for the difference between tumor biological variables and LNM variables was <0.001.

**Table 1 cancers-16-00659-t001:** Baseline characteristics of OTC, OPC, and HPC cases.

	OTC (*n* = 190)			HPV-Positive OPC (*n* = 118)		
No. (%)	ECS(+) (*n* = 113)	ECS(−) (*n* = 77)	*p*-value	ECS(+) (*n* = 72)	ECS(−) (*n* = 46)	*p*-value
Age (mean ± SD)	55.0 ± 14.9	55.7 ± 13.6	0.534	57.0 ± 9.4	55.9 ± 9.4	0.577
Sex (M:F)	62:51 (54.9:45.1)	46:31 (59.7: 40.3)	0.605	63:9 (87.5:12.5)	36:10 (78.3:21.7)	0.282
Tumor diameter (cm)	3.0	2.8	0.010	2.1	2.1	0.650
(median, [Q1–Q3])	[2.2–4.5]	[1.8–3.3]		[1.5–2.9]	[1.4–2.6]	
Tumor volume (cm^3^) (median, [Q1–Q3])	7.0 [2.0–11.6]	3.2 [1.4–6.4]	0.003	1.6 [0.5–4.8]	1.3 [0.4–3.8]	0.422
DOI (mm)	(*n* = 111)	(*n* = 76)	0.003	(*n* = 37)	(*n* = 20)	0.786
DOI ≤ 5	30 (27.0)	39 (51.3)		10 (24.4)	7 (29.2)	
5 < DOI ≤ 10	21 (18.9)	12 (15.8)		12 (29.3)	6 (25.0)	
10 < DOI	60 (54.1)	25 (32.9)		15 (36.6)	7 (29.2)	
T status	(*n* = 111)	(*n* = 76)	<0.001	(*n* = 69)	(*n* = 45)	0.720
T0				4 (5.8)	4 (8.9)	
T1	15 (13.5)	21 (27.6)		29 (42.0)	16 (35.6)	
T2	27 (24.3)	29 (38.2)		29 (42.0)	23 (51.1)	
T3	45 (40.5)	23 (30.3)		4 (5.8)	1 (2.2)	
T4	24 (21.6)	3 (3.9)		3 (4.3)	1 (2.2)	
LVI	(*n* = 110)	(*n* = 76)	0.028	(*n* = 58)	(*n* = 42)	0.057
Present	38 (34.5)	15 (19.7)		16 (27.6)	5 (11.9)	
PNI	(*n* = 109)	(*n* = 76)	0.017	(*n* = 60)	(*n* = 42)	0.078
Present	52 (47.7)	23 (30.3)		8 (13.3)	1 (2.4)	
WPOI	(*n* = 37)	(*n* = 25)	0.880			
Present	17 (45.9)	11 (44.0)				
Histologic grade	(*n* = 107)	(*n* = 76)	0.297	(*n* = 36)	(*n* = 21)	0.844
WD	39 (36.4)	36 (47.4)		1 (2.8)	0 (0.0)	
MD	48 (44.9)	30 (39.5)		27 (75.0)	15 (71.4)	
PD	20 (18.7)	10 (13.2)		8 (22.2)	6 (28.6)	
N status			<0.001			0.431
N1	23 (20.4)	46 (10.1)		15 (20.8)	11 (23.9)	
N2	77 (68.1)	31 (6.8)		51 (70.8)	34 (73.9)	
N3	13 (11.5)	0 (0.0)		6 (8.3)	1 (2.2)	
LNM number (median, [Q1–Q3])	2.0 [1.0–4.0]	1.0 [1.0–2.0]	<0.001	2.0 [1.0–4.0]	2.0 [1.0–3.0]	0.068
LNM largest node size (cm) (median, [Q1–Q3])	1.3 [0.9–1.9]	0.9 [0.4–1.4]	<0.001	3.0 [2.3–4.5]	2.5 [1.3–3.4]	0.008
	**HPV-negative OPC (*n* = 30)**			**HPC (*n* = 100)**		
No. (%)	ECS(+) (*n* = 17)	ECS(−) (*n* = 13)	*p*-value	ECS(+) (*n* = 81)	ECS(−) (*n* = 19)	*p*-value
Age (mean ± SD)	57.7 ± 10.1	60.6 ± 8.4	0.414	63.8 ± 9.0	65.2 ± 9.2	0.657
Sex (M:F)	16:1 (94.1:5.9)	12:1 (92.3:7.7)	0.999	79:2 (97.5:2.5)	17:2 (89.5:10.5)	0.336
Tumor diameter (cm)	2.9	2.5	0.395	3.0	2.7	0.692
(median, [Q1–Q3])	[2.2–3.5]	[1.6–2.9]		[2.2–4.5]	[2.2–4.1]	
Tumor volume (cm^3^) (median, [Q1–Q3])	6.5 [0.7–11.4]	3.2 [1.0–9.1]	0.702	4.5 [1.3–12.5]	4.4 [2.8–11.3]	0.983
DOI (mm)	(*n* = 11)	(*n* = 8)	0.589	(*n* = 33)	(*n* = 11)	0.622
DOI ≤ 5	2 (18.2)	2 (25.0)		11 (33.3)	4 (36.4)	
5 < DOI ≤ 10	4 (36.4)	1 (12.5)		11 (33.3)	5 (45.5)	
10 < DOI	5 (45.5)	5 (62.5)		11 (33.3)	2 (18.2)	
T status	(*n* = 17)	(*n* = 13)	0.004	(*n* = 81)	(*n* = 18)	0.570
T0						
T1	1 (5.9)	7 (53.8)		12 (14.8)	5 (27.8)	
T2	9 (52.9)	6 (46.2)		35 (43.2)	7 (38.9)	
T3	6 (35.3)	0 (0.0)		11 (13.6)	1 (5.6)	
T4	1 (5.9)	0 (0.0)		23 (28.4)	5 (27.8)	
LVI	(*n* = 13)	(*n* = 10)	0.402	(*n* = 65)	(*n* = 16)	0.006
Present	7 (53.8)	3 (30.0)		28 (43.1)	1 (6.3)	
PNI	(*n* = 14)	(*n* = 10)	0.341	(*n* = 65)	(*n* = 16)	0.171
Present	5 (35.7)	1 (10.0)		16 (24.6)	1 (6.3)	
WPOI						
Present						
Histologic grade	(*n* = 13)	(*n* = 9)	0.448	(*n* = 71)	(*n* = 16)	0.202
WD	1 (7.7)	0 (0.0)		9 (12.7)	0 (0.0)	
MD	8 (61.5)	8 (88.9)		50 (70.4)	15 (93.8)	
PD	4 (30.8)	1 (11.1)		12 (16.9)	1 (6.3)	
N status			0.002			<0.001
N1	1(5.9)	7 (53.8)		8 (9.9)	14 (73.7)	
N2	10 (58.8)	6 (46.2)		50 (61.7)	5 (26.3)	
N3	6 (35.3)	0 (0.0)		23 (28.4)	0 (0.0)	
LNM number (median, [Q1–Q3])	3.0 [1.0–6.5]	1.0 [1.0–3.0]	0.034	3.0 [2.0–5.0]	1.0 [1.0–2.0]	<0.001
LNM largest node size (cm) (median, [Q1–Q3))	2.1 [1.3–3.1)	1.3 [0.9–2.4]	0.156	2.5 [1.7–4.4]	1.4 [0.8–2.0]	0.002

ECS: extra-capsular spread, OTC: oral tongue cancer, HPV: human papillomavirus, OPC: oropharyngeal cancer, HPC: hypopharyngeal cancer, Q1–Q3: interquartile range, SD: standard deviation, DOI: depth of invasion, LVI: lymphovascular invasion, PNI: perineural invasion, WPOI: worst pattern of invasion, WD: well differentiated, MD: moderately differentiated, PD: poorly differentiated, LNM: lymph node metastasis.

**Table 2 cancers-16-00659-t002:** Univariable and multivariable analyses of risk factors for ECS in OTC.

	Univariable Analyses			Multivariable Analyses		
	OR	95% CI	*p*-Value	OR	95% CI	*p*-Value
Tumor dimension variables						
Tumor diameter (cm)	1.325	1.074–1.636	0.009			
Tumor volume (cm^3^)	1.058	1.015–1.103	0.008			
DOI (mm)						
DOI ≤ 5	1					
5 < DOI ≤ 10	2.275	0.968–5.345	0.059			
10 < DOI	3.119	1.602–6.075	<0.001			
T status						
T1	1			1		
T2	1.303	0.560–3.034	0.539	1.215	0.465–3.175	0.692
T3	2.739	1.193–6.291	0.018	1.905	0.732–4.953	0.186
T4	11.200	2.843–44.119	<0.001	6.483	1.544–27.225	0.011
Tumor biological variables						
LVI	2.146	1.078–4.269	0.030			
PNI	2.102	1.134–3.896	0.018			
WPOI	1.082	0.390–3.002	0.880			
Histologic grade						
WD	1					
MD	1.477	0.777–2.809	0.235			
PD	1.846	0.763–4.469	0.174			
LNM variables						
LNM number	1.681	1.292–2.186	<0.001			
LNM largest size (cm)	3.223	1.899–5.470	<0.001	2.288	1.328–3.942	0.003
N status						
N1	1			1		
N2	4.868	2.541–9.327	<0.001	3.192	1.549–6.581	0.002
N3	53.415	2.737–1042.499	0.009	35.870	1.808–711.619	0.019

ECS: extra-capsular spread, OTC: oral tongue cancer, OR: odds ratio, 95% CI: 95% confidence interval, DOI: depth of invasion, LVI: lymphovascular invasion, PNI: perineural invasion, WPOI: worst pattern of invasion, WD: well differentiated, MD: moderately differentiated, PD: poorly differentiated, LNM: lymph node metastasis.

**Table 3 cancers-16-00659-t003:** Univariable and multivariable analyses of risk factors for ECS in HPV-positive OPC.

	Univariable Analyses			Multivariable Analyses		
	OR	95% CI	*p*-Value	OR	95% CI	*p*-Value
Tumor dimension variables						
Tumor diameter (cm)	1.164	0.830–1.631	0.380			
Tumor volume (cm^3^)	1.050	0.970–1.136	0.225			
DOI (mm)						
DOI = 0	1					
0 < DOI ≤ 5	1.429	0.267–7.736	0.679			
5 < DOI ≤ 10	2.000	0.366–10.919	0.424			
10 < DOI	2.143	0.411–11.168	0.366			
T status						
T0	1					
T1	1.812	0.399–8.241	0.442			
T2	1.261	0.284–5.595	0.760			
T3	4.000	0.299–53.467	0.295			
T4	3.000	0.211–42.624	0.417			
Tumor biological variables						
LVI	2.819	0.941–8.445	0.064	1.553	1.123–2.149	0.008
PNI	6.308	0.758–0.088	0.088			
Histologic grade						
WD	1					
MD	0.591	0.006–57.099	0.822			
PD	0.436	0.004–45.507	0.726			
LNM variables						
LNM number	1.190	0.992–1.428	0.061			
LNM largest size (cm)	1.488	1.117–1.983	0.007	3.004	0.945–9.542	0.062
N status						
N1	1					
N2	1.108	0.455–2.697	0.822			
N3	3.215	0.414–25.03	0.264			

ECS: extra-capsular spread, OTC: oral tongue cancer, OR: odds ratio, 95% CI: 95% confidence interval, DOI: depth of invasion, LVI: lymphovascular invasion, PNI: perineural invasion, WD: well differentiated, MD: moderately differentiated, PD: poorly differentiated, LNM: lymph node metastasis.

**Table 4 cancers-16-00659-t004:** Univariable and multivariable analyses of risk factors for ECS in HPV-negative OPC.

	Univariable Analyses			Multivariable Analyses		
	OR	95% CI	*p*-Value	OR	95% CI	*p*-Value
Tumor dimension variables						
Tumor diameter (cm)	1.441	0.788–2.634	0.236			
Tumor volume (cm^3^)	1.046	0.959–1.142	0.312			
DOI (mm)						
DOI ≤ 5	1					
5 < DOI ≤ 10	4.000	0.211–75.658	0.355			
10 mm < DOI	1.000	0.098–10.166	0.999			
T status						
T1	1					
T2	7.307	0.872–61.222	0.067			
T3	64.991	1.739–2428.451	0.024			
T4	17.800	0.110–0.268	0.268			
Tumor biological variables						
LVI	2.722	0.479–15.463	0.259			
PNI	4.999	0.483–51.762	0.177			
Histologic grade						
WD	1					
MD	0.332	0.332–34.232	0.641			
PD	0.995	0.007–142.386	0.998			
LNM variables						
LNM number	1.648	0.963–2.822	0.068			
LNM largest size (cm)	1.463	0.799–2.681	0.218			
N status						
N1	1			1		
N2	8.077	0.974–66.965	0.053	8.077	0.974–66.965	0.053
N3	65.009	1.739–2429.666	0.024	65.009	1.739–2429.666	0.024

ECS: extra-capsular spread, OTC: oral tongue cancer, OR: odds ratio, 95% CI: 95% confidence interval, DOI: depth of invasion, LVI: lymphovascular invasion, PNI: perineural invasion, WD: well differentiated, MD: moderately differentiated, PD: poorly differentiated, LNM: lymph node metastasis.

**Table 5 cancers-16-00659-t005:** Univariable and multivariable analyses of risk factors for ECS in HPC.

	Univariable Analyses			Multivariable Analyses		
	OR	95% CI	*p*-Value	OR	95% CI	*p*-Value
Tumor dimension variables						
Tumor diameter (cm)	1.074	0.798–1.445	0.639			
Tumor volume (cm^3^)	1.023	0.978–1.069	0.320			
DOI (mm)						
DOI ≤ 5	1					
5 < DOI ≤ 10	0.800	0.168–3.799	0.779			
10 < DOI	2.000	0.302–13.265	0.473			
T status						
T1	1					
T2	2.083	0.565–7.672	0.270			
T3	3.373	0.434–26.196	0.245			
T4	1.880	0.464–7.617	0.377			
Tumor biological variables						
LVI	7.854	1.333–46.278	0.023			
PNI	3.444	0.562–21.119	0.181			
Histologic grade						
WD	1					
MD	0.172	0.008–3.628	0.258			
PD	0.439	0.014–14.163	0.642			
LNM variables						
LNM number	2.025	1.214–3.379	0.007			
LNM largest size (cm)	2.636	1.442–4.817	0.002	2.158	1.104–4.218	0.025
N status						
N1	1			1		
N2	15.663	4.532–54.136	<0.001	12.754	3.470–46.880	<0.001
N3	80.176	4.034–1593.502	0.004	35.989	1.617–775.018	0.022

ECS: extra-capsular spread, OTC: oral tongue cancer, OR: odds ratio, 95% CI: 95% confidence interval, DOI: depth of invasion, LVI: lymphovascular invasion, PNI: perineural invasion, WD: well differentiated, MD: moderately differentiated, PD: poorly differentiated, LNM: lymph node metastasis.

## Data Availability

The datasets generated and analyzed in the current study are available from the corresponding author upon reasonable request.

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
