# Peer review of "Extra-Capsular Spread of Lymph Node Metastasis in Oral, Oropharyngeal and Hypopharyngeal Cancer: A Comparative Subsite Analysis"

_cancers, 2024, doi:10.3390/cancers16030659_

Round 1

Reviewer 1 Report

Comments and Suggestions for Authors

The authors draw on extensive empirical evidence.Overall, this is a clear, concise, and well-written manuscript. The introduction is relevant and theory based. Sufficient information about the previous study findings is presented for readers to follow the present study rationale and procedures.

1. The main result addressed 23 oro- and hypo-pharyngeal cancer, neither primary tumor dimension nor T status were significant 24 factors for ECS occurrence. Site-dependent primary tumor or nodal status might contribute differ- 25 entially to the ECS development of LNM in HNC, but the baseline characteristics of study cohort in table 1 showed (LVI p value= 0.026), the authors should address these issue.

2. Each genotype should consider original or relevant for ECS occurrence, i think the specific gap in the field does not meet the paper address.

3. What is different from the paper “ Predictors of Extracapsular Spread in T1/T2 Oral Squamous Cell Carcinoma: A retrospective study ” and your study ?

4. Thre paper do not provide more specific knowledge for oro- and hypo-pharyngeal cancer.

5. Please describe how the conclusions are or are not consistent with the
Predictors of Extracapsular Spread in T1/T2 Oral Squamous Cell Carcinoma: A retrospective study

6. The authors miss many references.
7. All variable should not combined in the figures.

Author Response

Reviewer #1 Report

Comments and Suggestions for Authors

The authors draw on extensive empirical evidence. Overall, this is a clear, concise, and well-written manuscript. The introduction is relevant and theory-based. Sufficient information about the previous study findings is presented for readers to follow the present study rationale and procedures.

  1. The main result addressed oro- and hypo-pharyngeal cancers, neither primary tumor dimension nor T status were significant factors for ECS occurrence. Site-dependent primary tumor or nodal status might contribute differentially to the ECS development of LNM in HNC, but the baseline characteristics of the study cohort in Table 1 showed (LVI p value= 0.026), the authors should address these issues.

Answer: Thank you for your comment.

In our cohort, there were uneven findings between ECS(+) and ECS(-) cases, in OTC, OPC, and HPC. These findings suggested that some potential factors could contribute differentially to the ECS occurrence of LNM in HNC. Particularly, tumor dimensional variables, biological variables, and LNM variables were significantly different factors between ECS(+) and ECS(-) cases in OTC (Table 1). In HPV-positive OPC, the frequency of LVI, and LNM variables differed significantly between ECS(+) and ECS(-) cases. In addition, LNM variables were significant risk factors for ECS in HPV-negative OPC and HPC.

Based on these uneven distributions of some variables in HNC subsites, we conducted the univariable and multivariable analyses for ECS according to each subsite of HNC (Table 2-5). We found that LVI and LNM variables were significant risk factors for ECS in HPV-positive OPC (Table 3).

We clearly stated these differential risk factors for ECS according to the HNC subsites in the Results and Discussion section.

Results: In the case of HPV-positive OPC, no tumor dimensional variables were risk factors for ECS, while the largest node size of LNM was a risk factor for ECS in HPV-positive OPC (p <0.001). Among tumor biological variables, LVI was a risk factor for ECS in HPV-positive OPC (p <0.05) (Table 3).

Results: In HPV-positive OPC, models containing a tumor biological variable (LVI) and LNM variable (size of the largest LNM) predicted ECS with AUC values of 0.581 (95%CI = 0.512-0.649) and 0.754 (95%CI = 0.672-0.835), respectively.

  1. Each genotype should consider original or relevant for ECS occurrence, I think the specific gap in the field does not meet the paper address.

Answer: Thank you for your comments.

In this study, we explored the HNC subsite-specific effects on ECS occurrence; unfortunately, we did not investigate the tumor genotype and specific tumor biology in this study. All HNCs included in this study were squamous cell carcinomas. We only got some hints from our analyses that some tumor biological factors might affect ECS occurrence in HPV-positive OPC and HPV-negative OPC. However, the detailed information about tumor biology, genotype, and biomarkers and their potential impacts on ECS occurrence remained unanswered in this study. We mentioned these study limitations in the Discussion section.

Discussion:

Third, we did not analyze some biomarkers and mechanisms reported as predictors of ECS in earlier studies. Rather, we simply included pathological findings, which were readily available, under the category of tumor biological factors. Tumor biology dictated by specific molecular markers should be explored further in future studies.

  1. What is different from the paper “Predictors of Extracapsular Spread in T1/T2 Oral Squamous Cell Carcinoma: A Retrospective Study” and your study?

Answer: Thank you for your comments.

We reviewed this paper entitled [Predictors of Extracapsular Spread in T1/T2 Oral Squamous Cell Carcinoma: A Retrospective Study. J Oral Biol Craniofac Res. 2022 Jul-Aug; 12(4): 449–453.], again.

In this study, authors found that tumors of the oral tongue among oral SCC and PNI were independent risk factors for ENE (ECS) and primary tumor features, and LN size was significantly associated with ENE (ECS), which was essentially in agreement with our study.

However, several points were different from ours in terms of study design, and subjects.

The study by S. Tandon. et al. only included early-stage oral SCC (cT1-2N0) in their analysis; meanwhile, our study included major three subsites of HNC with all TNM stages, which enabled us to compare ECS risk factors comparatively. Also, we included N(+) diseases (not limited to N0) in our analysis, and we could investigate the status of LNM (number or size of LNM, N status) and the potential impact of LNM status on ECS. Therefore, our study was more comprehensive and comparative than the study by S. Tandon, et al..

  1. The paper does not provide more specific knowledge for oro- and hypo-pharyngeal cancer.

Answer: Thank you for your comments.

Indeed, there were not sufficient numbers of OPC and HPC (n = 176 and 117), compared to OTC (n = 570), to draw a more solid conclusion. Nevertheless, we could demonstrate that tumor dimensional variables were not significant determinants for ECS in OPC and HPC, in comparison to OTC. And, some tumor biological variables of primary tumor may affect the ECS occurrence in OPC. More importantly, LNM variables were significantly associated with ECS in OPC and HPC, as well as OTC. Our findings were supported by the clinical situation that many unknown primary tumors with neck metastasis originate from hidden OPC and HPC.

This is not a new finding or insight. However, we confirmed that ECS and LNM could be independent of primary tumor dimension (physical factors) in OPC and HPC, through a comparative analysis with OTC.

  1. Please describe how the conclusions are or are not consistent with the Predictors of Extracapsular Spread in T1/T2 Oral Squamous Cell Carcinoma: A Retrospective study

Answer: Thank you for your comments.

In the study by S. Tandon, et al., the authors found that tumors of the oral tongue among oral SCC and PNI were independent risk factors for ENE (ECS) and primary tumor features, and LN size was significantly associated with ENE (ECS).

In agreement with our study, they showed that tumor dimensional factors (DOI) and LNM factors (size) were important in ECS occurrence in early-stage oral cavity SCC. However, PNI was not a significant risk factor for ECS in OTC, in our study.

However, the study by S. Tandon. et al. only included early-stage oral SCC (cT1-2N0) in their analysis; meanwhile, our study included major three subsites of HNC with all TNM stages, which enabled us to compare ECS risk factors comparatively. Also, we included N(+) diseases (not limited to N0) in our analysis, and we could investigate the status of LNM (number or size of LNM, N status) and the potential impact of LNM status on ECS.

  1. The authors miss many references.

Answer: Thank you for your comments.

We conducted an extensive literature search again and added several relevant articles to our topic.

References [added]

  1. Alvi, A.; Johnson, J.T. Extracapsular spread in the clinically negative neck (N0): implications and outcome. Otolaryngology–Head and Neck Surgery 1996, 114, 65-70.
  2. Liao, C.-T.; Lee, L.-Y.; Hsueh, C.; Lin, C.-Y.; Fan, K.-H.; Wang, H.-M.; Hsieh, C.-H.; Ng, S.-H.; Lin, C.-H.; Tsao, C.-K. Pathological risk factors stratification in pN3b oral cavity squamous cell carcinoma: focus on the number of positive nodes and extranodal extension. Oral Oncology 2018, 86, 188-194.
  3. Puri, S.K.; Fan, C.-Y.; Hanna, E. Significance of extracapsular lymph node metastases in patients with head and neck squamous cell carcinoma. Current opinion in otolaryngology & head and neck surgery 2003, 11, 119-123.
  4. Suton, P.; Salaric, I.; Granic, M.; Mueller, D.; Luksic, I. Prognostic significance of extracapsular spread of lymph node metastasis from oral squamous cell carcinoma in the clinically negative neck. International journal of oral and maxillofacial surgery 2017, 46, 669-675.
  5. Brannan, A.G.; Johnstone, P.A.; Cooper, J. Extracapsular tumor extension in cervical lymph nodes: reconciling the literature and seer data. Head & neck 2011, 33, 525-528.
  6. Wenzel, S.; Sagowski, C.; Kehrl, W.; Metternich, F. The prognostic impact of metastatic pattern of lymph nodes in patients with oral and oropharyngeal squamous cell carcinomas. European Archives of Oto-Rhino-Laryngology and Head & Neck 2004, 261, 270-275.
  7. Tandon, S.; Bera, R.N.; Singh, A.K.; Mishra, M. Predictors of Extracapsular Spread in T1/T2 Oral Squamous Cell Carcinoma: A retrospective study. Journal of Oral Biology and Craniofacial Research 2022, 12, 449-453.
  8. Mamic, M.; Lucijanic, M.; Manojlovic, L.; Muller, D.; Suton, P.; Luksic, I. Prognostic significance of extranodal extension in oral cavity squamous cell carcinoma with occult neck metastases. International Journal of Oral and Maxillofacial Surgery 2021, 50, 309-315.
  9. Su, Z.; Duan, Z.; Pan, W.; Wu, C.; Jia, Y.; Han, B.; Li, C. Predicting extracapsular spread of head and neck cancers using different imaging techniques: a systematic review and meta-analysis. International journal of oral and maxillofacial surgery 2016, 45, 413-421.
  10. Lewis Jr, J.S.; Carpenter, D.H.; Thorstad, W.L.; Zhang, Q.; Haughey, B.H. Extracapsular extension is a poor predictor of disease recurrence in surgically treated oropharyngeal squamous cell carcinoma. Modern Pathology 2011, 24, 1413-1420.
  11. Greenberg, J.S.; Fowler, R.; Gomez, J.; Mo, V.; Roberts, D.; El Naggar, A.K.; Myers, J.N. Extent of extracapsular spread: a critical prognosticator in oral tongue cancer. Cancer: Interdisciplinary International Journal of the American Cancer Society 2003, 97, 1464-1470.
  12. Wreesmann, V.B.; Katabi, N.; Palmer, F.L.; Montero, P.H.; Migliacci, J.C.; Goenen, M.; Carlson, D.; Ganly, I.; Shah, J.P.; Ghossein, R. Influence of extracapsular nodal spread extent on prognosis of oral squamous cell carcinoma. Head & neck 2016, 38, E1192-E1199.
  13. Agarwal, J.P.; Kane, S.; Ghosh‐Laskar, S.; Pilar, A.; Manik, V.; Oza, N.; Wagle, P.; Gupta, T.; Budrukkar, A.; Murthy, V. Extranodal extension in resected oral cavity squamous cell carcinoma: more to it than meets the eye. The Laryngoscope 2019, 129, 1130-1136.

  1. All variables should not be combined in the figures.

Answer: Thank you for your comments.

In these analyses (ROC curve analyses to predict ECS), we included significant variables among tumor dimensional, biological, and LNM variables to plot the ROC curve for the prediction of ECS in each HNC subsite. Then, we also inserted the ROC curve using all significant variables combined, to see any synergistic (additive) effects for the prediction of ECS using all significant variables. Therefore, we decided to leave the figures (ROC curves) as they were.

[END]

Reviewer 2 Report

Comments and Suggestions for Authors

Chapter 2.2. Histopathology Analysis

Line 109: ...tumour staging was based on the AJCC 7th edition. Please, consider re-staging to AJCC 8th edition.

Chapter 2.4. Statistical Analysis

line 137: ...We included statistically significant variables in the univariable model in multivariable models.  It seems to be, that you used p<0.05. In this situation, p<0.25 is most commonly used cut off for variable to continue to multivariable analysis. Please, specify the p-value.

Results - Table 1

You classified N0 in 380 patients with OTC, in 20 patients with HPV-positive OPC, in 8 with HPV-negative OPC and in 17 patients with hypopharyngeal cancer.

Do I understand correctly that patients with no LN involvement at all were included into the analysis as ECS- patients?

If I am correct, it would be a source of massive bias, N0 patient must be excluded.

I suggest to include into the analysis only patients with involved LN - it would be 190 patients with OTC (113 ECS+ and 77 ECS-) , 118 patients with HPV+ OPC (72 ECS+ and 46 ECS-), 30 patients with HPV- OPC (17 ECS+ and 13 ECS-) and 100 patients with HPC (81 ECS+ and 19 ECS-).

Author Response

Comments and Suggestions for Authors

Chapter 2.2. Histopathology Analysis

Line 109: ...tumour staging was based on the AJCC 7th edition. Please, consider re-staging to AJCC 8th edition.

Answer: Thank you for your comments.

As you know, the presence or absence of ECS is incorporated into the criteria for N classification in the AJCC 8th edition. Meanwhile, the number, size, or laterality of nodal involvement was the criteria for N classification in the AJCC 7th edition (irrespective of ECS).

In this study, we aimed to compare N burden (status) according to the presence or absence of ECS (independently of ECS status). Therefore, we adopted AJCC staging system of 7th edition, instead of 8th edition.

Chapter 2.4. Statistical Analysis

line 137: ...We included statistically significant variables in the univariable model in multivariable models. It seems to be, that you used p<0.05. In this situation, p<0.25 is most commonly used cut off for variable to continue to multivariable analysis. Please, specify the p-value.

Answer: Thank you for your comments.

We used a stepwise variable selection method for multivariable analysis rather than variable selection using specific p values. We corrected the text, accordingly.

Statistical analysis:

Univariable and multivariable logistic regression analyses were used to identify risk factors of ECS, according to primary tumor subsite. We designed multivariable models from the univariable analyses using a stepwise variable selection method.

Results - Table 1

You classified N0 in 380 patients with OTC, in 20 patients with HPV-positive OPC, in 8 with HPV-negative OPC, and in 17 patients with hypopharyngeal cancer.

Do I understand correctly that patients with no LN involvement at all were included into the analysis as ECS- patients?

If I am correct, it would be a source of massive bias, N0 patients must be excluded.

I suggest including into the analysis only patients with involved LN - it would be 190 patients with OTC (113 ECS+ and 77 ECS-), 118 patients with HPV+ OPC (72 ECS+ and 46 ECS-), 30 patients with HPV- OPC (17 ECS+ and 13 ECS-) and 100 patients with HPC (81 ECS+ and 19 ECS-).

Answer: Thank you for your comments.

Following the review opinion, we excluded the N0 cases in our analysis and reanalyzed the significance of ECS in N(+) cases.

Corrections

Study patients:

Among 863 cases, we only included N(+) cases (n = 438) for our ECS analyses, excluding 425 N0 cases (190 in OTC, 118 in HPV-positive OPC, 30 in HPV-negative OPC, and 100 in HPC).

We also revised Table 1-5, and Figure 2.

Revised Table 1.

Revised Table 2-5.

Revised Figure 2.

[END]

Reviewer 3 Report

Comments and Suggestions for Authors

Authors based on a retrospective study of several hundreds of head and neck cancer patients concluded that the size of the tongue carcinoma is related to extra capsular spread (ECS) of lymph node metastasis. On the contrary the size of oro and hypo pharyngeal cancer, were not related to  ECS.

This study is relatively well justified since there are some discrepancies on the literature on the issue. It is also interesting since there are relatively recent studies showing different “behaviour” of the various head and neck carcinomas (e.g. unknown primary vs laryngeal).

Their data are explicitly presented in tables and although I do not have the statistical knowledge to examine their multivariate and roc analyses, it is clear to the reader that tumours of different sites and different sizes seem to behave differently e.g. T2 tongue cancers tend to spread extracapsularly more than T2 hypopharengeal cancers. Actually, this is an expected finding based on previous publications of the authors and other researchers. The same is true for their finding that ECS is related to lymph nodal metastasis. Nevertheless, their data is valuable and clearly presented in tables and provides the conclusion that we should distinguish the various head and neck cancers and not study them all together. Their site is an important distinction as it is for example their hpv status the last several years.

In conclusion, it is a study based on classical and well-known markers, such as the size of the tumor and lymph node metastasis, thus it does not offer novel information based on biomarkers. Although it is interesting, the authors do not justify clearly its practical value and how their findings can be useful for an upcoming research e.g. on biomarkers of ECS         

Author Response

Reviewer #3 Report

Comments and Suggestions for Authors

Authors based on a retrospective study of several hundreds of head and neck cancer patients concluded that the size of the tongue carcinoma is related to extra-capsular spread (ECS) of lymph node metastasis. On the contrary, the size of oro and hypo-pharyngeal cancer were not related to ECS.

This study is relatively well justified since there are some discrepancies on the literature on the issue. It is also interesting since there are relatively recent studies showing different “behaviour” of the various head and neck carcinomas (e.g. unknown primary vs laryngeal).

Their data are explicitly presented in tables and although I do not have the statistical knowledge to examine their multivariate and roc analyses, it is clear to the reader that tumours of different sites and different sizes seem to behave differently e.g. T2 tongue cancers tend to spread extracapsularly more than T2 hypopharyngeal cancers. Actually, this is an expected finding based on previous publications of the authors and other researchers. The same is true for their finding that ECS is related to lymph nodal metastasis.

Nevertheless, their data is valuable and clearly presented in tables and provides the conclusion that we should distinguish the various head and neck cancers and not study them all together. Their site is an important distinction as it is for example their hpv status the last several years.

In conclusion, it is a study based on classical and well-known markers, such as the size of the tumor and lymph node metastasis, thus it does not offer novel information based on biomarkers. Although it is interesting, the authors do not justify clearly its practical value and how their findings can be useful for upcoming research e.g. on biomarkers of ECS.

Answer: Thank you for your comprehensive review and comments.

ECS, which is defined as cancer cell extension beyond the lymph node capsule, is a major predictor of recurrence, metastasis and poor prognosis in HNC patients, and treatment should be intensified in patients with ECS-positive tumors. Therefore, under-standing the risk factors and sequence of ECS occurrence may help guide treatment options and predict prognosis in HNC patients.

One of the study limitations was that we did not analyze some biomarkers and mechanisms reported as predictors of ECS in earlier studies. Rather, we simply included pathological findings, which were readily available, under the category of tumor biological factors. Tumor biology dictated by specific molecular markers should be explored further in future studies.

Regarding the lymphatic progression and spread of HNC, the mechanisms underlying ECS development have been investigated in several studies, but the relative significance of dimensional (physical) versus biological factors remains unclear. Many previous studies have focused on the differential expression of various genetic, epigenetic, or molecular markers in primary tumors with and without ECS, but only a few studies have directly compared ECS tumors with intra-nodal metastasis.

A strong correlation between ECS of LNM and poor clinical outcomes in cancer patients can be explained by the aggressive phenotype of the primary tumor, and ECS findings may be just one manifestation of this aggressive phenotype. Indeed, the hypothesis that molecular determinants of ECS are present in the primary tumor is supported by recent studies.

Meanwhile, several studies have investigated the role of tumor microenvironment in ECS tumor formation and suggested that phenotypic changes occur from intra-nodal metastasis to ECS, rather than intrinsic properties of the primary tumor.

If the molecular determinants of ECS are solely dependent on the primary tumor characteristics, the frequency of ECS will be similar (evenly distributed) regardless of the burden of LNM, or reciprocal (early manifestation of ECS even in N1 disease). In contrast, we demonstrated that ECS occurrence is related to LNM progression in this study. Although this does not completely exclude the possibility that ECS determinants are already present in the primary tumor, molecular changes during LNM progression could be critical for ECS formation and dissemination, in addition to inherent primary tumor factors. Therefore, therapeutic interventions for ECS-prone changes or reversal of ECS-related changes during LNM progression may be a promising treatment option for improving the dismal ECS-associated outcomes in HNC patients.

These points should be further investigated in the following research for clinical application, and our study could provide background data about the future ECS study.

Thank you for your valuable comments.

[END]